# Development & validation of a health awareness booklet: "Reproductive health schemes for tribal women of Jharkhand" a study protocol

Rohit Raj[1], Jarina Begum[1]*, Syed Irfan Ali[1‡], Manasee Panda[2‡], Vani Lakshmi R[3‡]

1 Department of Community Medicine, Manipal Tata Medical College, Manipal Academy of Higher Education, Manipal, India, 2 Department of Community Medicine, Bhima Bhoi Medical College, Balangir, Odisha, India, 3 Department of Health Technology and Informatics, Prasanna School of Public Health, Manipal Academy of Higher Education, Manipal, India

◉ These authors contributed equally to this work.
‡ SIA, MP and VLR also contributed equally to this work.
* jarina.begum@manipal.edu

## Abstract

In India, women of reproductive age constitute of 22.2% of the total population. In tribal areas there are disparity in maternal health care utilization, lack of accessibility to safe menstrual health, malnutrition, and low contraceptive usage. This research will aim to study the utilization of healthcare schemes among tribal reproductive-age groups in Jharkhand. Based on the findings a health awareness booklet will be developed and validated. This study can contribute in designing effective interventions and policies that cater to the specific needs of tribal communities, ultimately improving their access to quality healthcare services. The proposed research will follow the sequential explanatory approach, conducted in two phases. Phase one will assess the knowledge, attitude, and practices of reproductive healthcare schemes among tribal women. We will also explore the facilitators and challenges in the provision of schemes from different stakeholder perspectives. Phase two the development and validation of the awareness booklet for the tribal women will be done. The sampling frame will consist of adolescent girls (10–19 years) and reproductive-age group women (15–49 years), from the tribal communities in East Singhbhum district of Jharkhand. Qualitative aspect will include study participants such as Block Program Manager, Block Trainer, Accredited Social Health Activist (ASHA), Auxiliary Nurse Midwifery (ANM), Aanganwadi Workers (AWW), Medical Officer, Village leaders, and Tribal Women. Research on the knowledge, attitude, and practices towards health coverage schemes in tribal regions of India is essential for addressing health disparities. Improving awareness and understanding, overcoming attitudinal barriers, informing policy development, and promoting equity and social justice in healthcare are

**Data availability statement:** As this manuscript is a study protocol. For this no results are reported. A deidentified research data will be made publicly available when the study is completed and published.

**Funding:** The author(s) received no specific funding for this work.

**Competing interests:** The authors have declared that no competing interests exist.

**Abbreviations:** ASHA, Accredited Social Health Activist; ANM, Auxiliary Nurse Midwifery; AWW, Aanganwadi Workers; ANC, Antenatal Care; CVI, Content Validity Index; CTRI, Clinical Trial Registry-India; IEC, Institute Ethics Committee; ISC, Institutional Scientific Committee; KAP, Knowledge Attitude Practices; NFHS, National Family Health Survey; MOIC, Medical Officer Incharge; PIS, Participatory Information Sheet; PNC, Postnatal Care; SBA, Skilled Birth Attendants; SRS, Sample Register System; SWOC, Strength Weakness Opportunities Challenges; KAP, Knowledge Attitude Practices

some of the disparities. This study will help to increase in knowledge, awareness, and attitude for reproductive age group tribal women health coverage schemes.

## Trial registration

Clinical Trial Registry-India (CTRI) CTRI/2024/03/063824

## Introduction

Women's reproductive health is a significant issue in India. Women of reproductive age constitute 22.2% of the total population. To ensure the population's growth and development, the country has a lot of work to perform. Many studies on women's health coverage programs in India have been done to understand the problems. Over the past 20 years, India has experienced increased economic growth, but it has performed poorly in terms of health and human development indexes. Access to healthcare and significant health inequities have become worse and persisted across states' communities. In tribal areas, there are disparities in maternal health care utilization, lack of accessibility to safe menstrual health, malnutrition, and low in contraceptive usage.

The maternal mortality ratio of Jharkhand state showed a declining trend (56 per lakh live births) compared to India (97 per lakh live births) as per Sample Registration System (SRS): 2018–20 [1]. However, the use of *"Maternal Child Health services"* in Jharkhand is lagging. National Family Health Survey (NFHS)-5 data showed the percentage of mothers who had at least four Antenatal Care (ANC) visits is 54.2 in India, whereas it is only 36.4% in Jharkhand. Similarly, rural Jharkhand has 92.3% of registered pregnancies, but only 36.4% had four ANC visits, 13.2% of mothers consuming "Iron Folic Acid" for 180 days (about six months), at the time they were pregnant, and 66.7% *of* mothers received postnatal care[2]. In terms of lack of accessibility to safe menstrual health, a study context to the rural part of West Singhbhum found that 38% of the adolescent girls were anemic, and 27% of the adolescent girls use local napkins during their menstruation [2]. Anemia in pregnant women (15–49 years old) is 59.2% and in adolescent girls (15–19 years) is 66.7% respectively according to the Jharkhand NFHS 5 report, this is slightly higher from the Indian context with 54.3% and 60.2% respectively [3]. Whereas it was found that rural women with Body Mass Index below normal was 29.2% as compared to the overall India context with 21.2%. The percentage of male sterilization is 0.2% in rural areas of Jharkhand followed by female sterilization at 0.4%, condom usage at 3.5%, and pill usage at 3.1% only. Healthcare providers discussed family planning with non-user females. in rural areas of Jharkhand is 29.5% reported from NFHS 5 report [3].

On literature search various evidence were reported regarding the reproductive healthcare service disparities among the tribal communities of India. As per one of the survey carried out in India discovered the considerable caste/tribe disparities

among the use of "Maternal Child Health care services" in the chosen Indian states. The main causes for the under-utilization of "these services", particularly, for members of marginalized social groups, having restricted availability & the absence of comprehensive healthcare [4]. Another study analyzed the level of disparity in maternal health care, including complete antenatal care (full ANC), Skilled Birth Attendants (SBA), and Postnatal Care (PNC) in rural India [5]. A qualitative study was done in Uttar Pradesh regarding "Accessibility of Reproductive Health Related Schemes for Pregnant and Lactating Rural Women", it was found that the study focused into the way women experienced access and hurdles in using health care schemes [6]. Another study was conducted in the three Indian states of "*Madhya Pradesh", "Jharkhand", and "Chhattisgarh"*, which are inhabited by tribes. The main aim of the current study was to analyze the variables influencing the use of maternal health services. The "National Family Health Survey" data were taken in the study. According to the study findings, there were larger "socio-economic" disparities in the use of MCH services in each of the three states that were the subject of the investigation. As per the "regression" analysis, people living in the tribal area are less likely than other caste groups to use maternal and child health care facilities and benefits [7]. Therefore, a study stated that one of India's most fragile and impoverished states is Jharkhand. There aren't many community-based programs and campaigns that prioritize adolescent and youth public health issues like gender and sexuality [8].

However, there are some major health coverage schemes running in India to improve the health of a reproductive age group women, such as Janani Suraksha Yojana, Janani Sishu Swasthya Karyakram, Pradhan Mantri Matri Vandana Yojana, Rashtriya Kishori Swasthya Karyakram, Poshan Abhiyan, Anemia Mukt Bharat.

Hence this research will aim to study the utilization of healthcare schemes among tribal reproductive-age groups in Jharkhand. Based on the findings a health awareness booklet will be developed and validated. Our study can contribute to designing effective interventions and policies that cater to the specific needs of tribal communities, ultimately improving their access to quality healthcare services. Research on the knowledge, attitude, and practices towards health coverage schemes in tribal regions of India is essential for addressing health disparities. Improving awareness and understanding, overcoming attitudinal barriers, informing policy development, and promoting equity and social justice in healthcare are some of the disparities. It can contribute to designing effective interventions and policies that cater to the specific needs of tribal communities, ultimately improving their access to quality healthcare services.

This research will aim to study the utilization of healthcare schemes among tribal reproductive-age groups in Jharkhand. Based on the findings a health awareness booklet will be developed and validated. Our study can contribute in designing effective interventions and policies that cater to the specific needs of tribal communities, ultimately improving their access to quality healthcare services. Therefore, this study will help to increase in knowledge, awareness, and attitude for reproductive age group tribal women health coverage schemes in the regional language with more pictorial content that can be understood easily.

## Objectives

### Phase 1: Need assessment.

1. To assess the knowledge, attitude, and practices of healthcare schemes among tribal women in the reproductive age group.

2. To understand facilitators and challenges in provision of tribal reproductive age group women healthcare schemes from different stakeholder perspectives.

### Phase 2: Development & Validation.

3. To develop & validate the awareness booklet for the tribal women of reproductive age group and evaluate the perception of participants towards it.

## Methods/design

We will follow the research design as "Sequential Explanatory Design" [9]. Leading to the generation of evidence for the development of a validated health awareness booklet for tribal women. This study protocol is a part of a Ph.D. project, and the study protocol was prepared and approved from the Institutional Scientific Committee (ISC) on 16th August 2023, followed by the approval from Institute Ethics Committee (IEC) with IEC Number MTMC/IEC/2023/53and the University.

The starting of the recruitment period for this study is as follows: the data collection under phase 1, Objective 1(Quantitative part) was started from 2nd March 2024 and ended on 2nd September 2024. Further, followed by Objective 2 (Qualitative aspect) and under phase 2, Objective 3: Development and Validation of the health awareness booklet for tribal women is planned to complete by 30th October 2026. The timeline for the research protocol will be three years as illustrated in (Fig 1).

### Study setting

The study will be conducted among tribal women in the reproductive age group (15–49 years) and tribal adolescent girls (10–19 years) residing in the field practice area of the Department of Community Medicine. The population comprises individuals from various tribal communities, including Santhal, Ho, Munda, Bhumij, Kharia, and Sabar tribes. The predominant languages spoken in the region are Santhali, Ho, Mundari, Bangla, and Hindi, reflecting the rich linguistic diversity of the area. The primary livelihood of the population includes agriculture, daily wage labor, and homemaking, which are influenced by the socio-economic and cultural fabric of the tribal communities. Geographically, the region is characterized by a mix of plateau, plain, and forested terrain, shaping the lifestyle and accessibility to health services.

### Sampling

East Singhbhum is a district located in the state of Jharkhand. It consists of two sub-divisions, Dalbhum and Ghatsila. Comprising of 11 blocks, Potka, Golmuri cum Jugsalai, Patamda, Ghatsila, Boram, Musabani, Dumaria, Bharagora,

| | STUDY PERIOD | | | | | |
|---|---|---|---|---|---|---|
| TIMEPOINT** | 0-6months | 6-12 | 12-18 | 18-24 | 24-30 | 30-36 |
| **ENROLMENT:** | | | | | | |
| **Eligibility screen** | X | | | | | |
| **Informed consent** | X | | | | | |
| **Data collection objective 1 and 2** | | X | X | | | |
| **Module Preparation** | | | | X | X | |
| **Booklet Validation and Piloting** | | | | | | X |
| | | | | | | |

**Fig 1. Timeline of the completion of work based on SPIRIT guidelines.**

Dalbhumgarh, Chakulia, and Gurbandha. A total of twelve villages will be taken from the two randomly selected blocks. The sampling frame for objective one will consists of the list of adolescent girls and reproductive age group women. For the qualitative aspect objective two, the sampling frame will include other stakeholders such as the Block Program Manager/ Officer, Block Trainer, Accredited Social Health Activist (ASHA), Auxiliary Nurse Midwifery (ANM), Aanganwadi Workers (AWW), Medical Officer, Village head, and the Tribal women of reproductive age group. Statistical analysis for objective 1 and 3 (feedback form), will be descriptive data analysis and chi-square test to know the association between two variables. Whereas, for objective 2, qualitative data thematic analysis will be done based on codes generated from the transcripts. In objective 1 the study design will be cross-sectional study. Population covered will be the adolescent girls (10–19 years) and reproductive age group (15–49 years) tribal women belonging to following tribes: Santhal, Ho, Munda, Bhumiz, Kharia, and Sabar. Following religion such as Sarna Dharam, Christianity, or Hinduism. Study Participants who will give consent to participate in the study. Migrant population and tribal women suffering from any mental illness will be excluded from the study.

The sample size was calculated through an online tool calculator.net keeping the level of confidence (z) at 95%, and marginal error(E)at 5%, population proportion 50% and population size 258454 [10]. Considering the drop out of 20%, with design effect of 1.5, the sample size becomes 385 (385 + 337) = 722 rounding up to 745. Formula used: n= $(Z_{1-\alpha/2})^{2}$ $*$ p(1-p)/d$^2$. Where n = sample size, Z = level of confidence, E = marginal error, P = Population proportion [11].

Simple random sampling and stratified sampling technique will be used. Out of two blocks, twelve villages will be chosen at random, six village from each block. From each village there will be stratified into two strata, tribal adolescent girls (n = 32/village) and tribal women of reproductive age group. (n = 30 per village). Sampling technique is depicted in form of flow chart mentioned in (Fig 2).

A Self-administered questionnaire will be used, and in case of illiterate study participants, questionnaire form will be filled by the interviewer/ or the person who will be going to collect the data, on behalf of study participant. Validation of the developed questionnaire will be through the external subject experts and finding the Content Validity Index (CVI) score. Reliability check of the questionnaire will be done through piloting among 10 percent of the study participants from the sample size of the main study and Cronbach's alpha score will be derived. Domains of the questionnaire will comprise of two sections, Section A- socio-demographic details and section B- Baseline survey consisting of questions on knowledge, attitude, and practices of tribal women towards relevant eight reproductive healthcare schemes of India.

Details of the eight reproductive healthcare schemes included are mentioned in Table 1. Participatory Information Sheet (PIS) will be explained, and Informed Consent (IC) will be taken from the study participants. Data will be collected based on the questionnaire. A total of 20–30 minutes will be given to the study participants and a maximum of 10–15 days per village will be given to collect the data. Quantitative data analysis in terms of frequency, percentage, proportion and for categorical data chi square test will be done. KAP score will be calculated and based on the median and interquartile range which will be finally interpret as low, moderate and high KAP score. All quantitative analysis will be through Jamovi software (Version solid 2.2.5).

For Objective two the research design included a Phenomenological approach, through this study participant can express there experienced field perspectives related to facilitators and challenges faced. The study population will cover tribal women of reproductive age group (1/village) and adolescent girl(1/village), Block program manager/officer (1/ block), block trainer (1/block), Accredited Social Health Activist (1/village), Auxiliary Nurse Midwifery (1/village), Aanganwadi Workers (1 per village), Medical Officer (1/block), and Village Leader (1/village). Purposive sampling will be done. 5–8 interviews per village will be done till the saturation of data over one year to collect and analyze it further. Interviewer's guide comprising of open-ended questions will be used for an in-depth interview with the broad domains including Demographic details. In-depth interview guide will be based on the Strength, Weakness, Opportunities, and Challenges (SWOC) faced in provision of tribal women.

The researcher will go to the assigned community settings after seeking permission from the State tribal Welfare Commissioner, District Surgeon, Deputy Commissioner, and from Medical Officer In-charge (MOIC). Qualitative data will

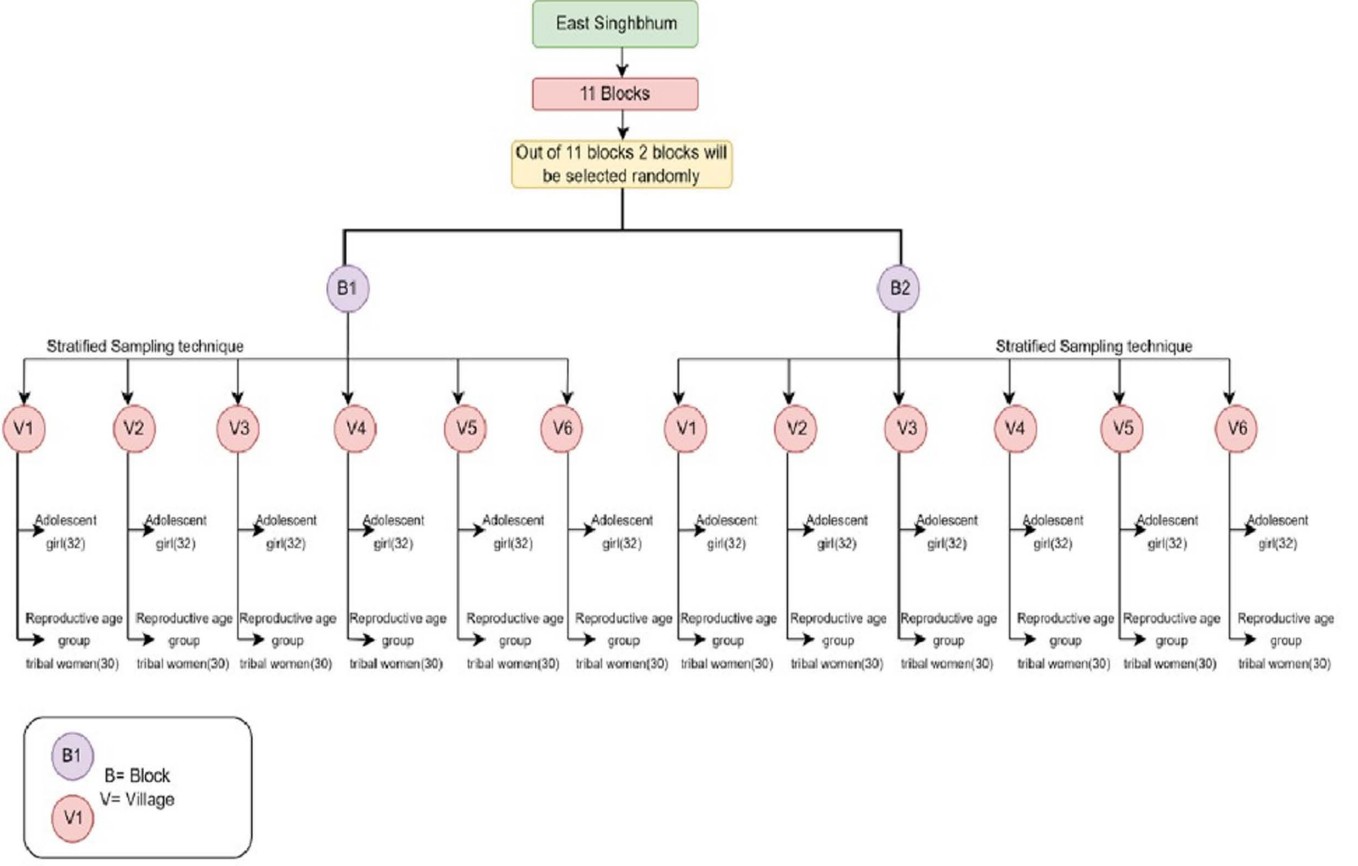

**Fig 2. Simple random sampling and stratified sampling technique.**

be collected by In-depth interviews of the study population after explaining the Participatory Information Sheet (PIS) and undertaking the signatures in the Consent forms. Approximately, 20–30 minutes will be given for the interview. Thematic analysis will be done through Atlas ti Software (Version 8).

Phase 2 will be the Development and Validation phase. Based on the results of survey among tribal women and in-depth interview of all the stake holders, a health awareness booklet will be developed on various schemes targeted for adolescent & reproductive age group tribal women mentioned in Table 1. The booklet will contain the information such

**Table 1. Healthcare schemes for reproductive age group tribal women in India.**

| Schemes for Adolescent Girls (10–19 Years) | Schemes for reproductive age group women (15–49 Years) |
|---|---|
| 1.Rashtriya Kishori Swasthya Karyakram | 1.Janani Suraksha Yojana |
| 2.POSHAN Abhiyan | 2.Janani Sishu Suraksha Karyakaram |
| 3.Anemia Mukt Bharat | 3.Pradhan Mantri Matru Vandana Yojana |
| | 4.Pradhan Mantri Surakshit Matritva Abhiyan |
| | 5.Family Planning Schemes under NHM including:<br>a. ASHA Scheme<br>b. Enhanced Compensatory Scheme |
| | Anemia Mukt Bharat |

as benefits and procedures to avail the incentives along with the strategies to overcome the challenges in utilization of services in the eight health schemes selected for the study. Booklet will be validated by the subject experts and piloted among the study participants. Booklet will also discuss about the clinical features covering sign symptoms, diagnosis, and treatment modalities available for any health-related events, the benefits along with the consequences of non-utilization of services benefits of the schemes will also cover the impact of the related events, the benefits along with the consequences of non-utilization of services in terms of adverse health outcomes.

Validation of the booklet will be done with the help of a group of experts (6–10) and by piloting among the study participants. Content validity tools will be used to validate the booklet with the Item generation (through existing literature, previous research, expert knowledge), expert panel review with the help of subject experts, health professional, medical education experts and researcher will review the item, and the rate/rank will be allotted based on their judgement, item selection and modification by refining and revision based on expert comments, and Content Validity Index (CVI) Calculation will be done. Piloting of the booklet developed will be done in the study population (reproductive age tribal women and adolescent girls). Around 20% of the study participants will be included from the main sample size 745. Total 149 rounding up to 150, 12–13 participants per village from 12 villages will be taken. Hence the sampling method used here will be stratified sampling technique and simple random sampling. A series of training sessions (one per each village each of two hours session) will be conducted for the adolescent girls and reproductive age women. Training sessions will be through interactive sessions, Nukkad Natak (street play) along with Audio/Visual clips of the booklet in the local language. The sessions will include briefing on the content, utilization, challenges and ways to address the healthcare schemes mentioned in the booklet. Training and awareness sessions will be delivered on monthly routine immunization day, weekly morning Gram Sabha, and weekly Mahila Mandal meetings.

Training will be delivered in Hindi and Santhali (tribal) language, other tribal language such as Ho and Mundari can be incorporated as per need and demand from the study participants.

Evaluation of the feedback of participants towards the awareness booklet through a feedback form will be done after the training sessions. Feedback after training will cover the questions related to the training session, time, pace, and satisfaction of the participants. Whereas feedback for booklet will cover the content appropriateness, satisfaction, understanding the content, and effectiveness of local language. Table 1 below, represents the name of the eight healthcare schemes of India that will be included in the study.

## Ethical aspects

This protocol was first approved by the Institutional Scientific Committee (ISC) completed with proposal no: F-2023-20, on 16th August 2023, followed by the Institute Ethics Committee (IEC) on 16 October 2023 till 30 October 2026. Ethical clearance has been done from Institutional Ethics Committee No: MTMC/IEC/2023/53. Permission from the Jharkhand State Tribal Welfare Commissioner, District Civil Surgeon, and gatekeepers (village leaders) were taken. Participation Information Sheet will be explained to the study participants, and written Informed consent will be taken from the reproductive age group tribal women and adolescent girls in presence of the witness before data collection. In case of minor's parent or guardian written assent/consent will be taken prior to the data collection. The thumb impression will be taken in case of illiterate study participant's. This study is registered at Clinical Trial Registry-India (CTRI) with CTRI No: CTRI/2024/03/063824 on March 8, 2024. Confidentiality of study participants will be maintained. Name and designation will be coded and will not be exposed.

## Discussion

Through this study the baseline data (KAP) of health care schemes utilization among tribal women of reproductive age group will be reported also this study will Identify the challenges in the utilization of services from different stakeholder perspectives. Development and Validation of the booklet covering the reproductive healthcare schemes targeted for tribal

women will help in increase knowledge among tribal women after the training program on various health schemes (8) mentioned in the booklet. This research would provide valuable insights for policymakers, healthcare providers, and implementers to tailor the programs to the specific needs and preferences of the tribal communities, ensuring their effective utilization and impact. Community at large may benefit from the intervention and recommendations of this study based on the findings. Hence, this study can act as a baseline study based on which some larger studies may be planned to get more generalizable findings.

## Supporting information

**S1 File. MTMC new protocol.**
(DOCX)

**S1 Checklist. SPIRIT-Outcomes 2022 Checklist (for combined completion of SPIRIT 2013 and SPIRITOutcomes 2022 items).**
(PDF)

## Author contributions

**Conceptualization:** Rohit Raj, Jarina Begum, Syed Irfan Ali, Manasee Panda, Vani Lakshmi R.

**Formal analysis:** Rohit Raj, Jarina Begum, Manasee Panda.

**Methodology:** Rohit Raj, Jarina Begum, Syed Irfan Ali, Vani Lakshmi R.

**Resources:** Rohit Raj.

**Software:** Vani Lakshmi R.

**Supervision:** Jarina Begum, Syed Irfan Ali, Manasee Panda, Vani Lakshmi R.

**Validation:** Rohit Raj, Jarina Begum, Syed Irfan Ali, Manasee Panda, Vani Lakshmi R.

**Visualization:** Rohit Raj, Jarina Begum, Syed Irfan Ali, Manasee Panda.

**Writing – original draft:** Rohit Raj, Jarina Begum.

**Writing – review & editing:** Jarina Begum, Syed Irfan Ali, Manasee Panda, Vani Lakshmi R.

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
