## [Decision Letter · Decision Letter 0]

20 Dec 2024

PONE-D-24-37230Development & Validation of a health awareness booklet: “Reproductive Health Schemes for tribal women of Jharkhand” a study protocolPLOS ONE

Dear Dr. Begum

Thank you for submitting your manuscript to PLOS ONE. After careful consideration, we feel that it has merit but does not fully meet PLOS ONE’s publication criteria as it currently stands. Therefore, we invite you to submit a revised version of the manuscript that addresses the points raised during the review process.

We look forward to receiving your revised manuscript.

Kind regards,

George Kuryan

Academic Editor

PLOS ONE

Journal Requirements:

3. We note that the original protocol that you have uploaded as a Supporting Information file contains an institutional logo. As this logo is likely copyrighted, we ask that you please remove it from this file and upload an updated version upon resubmission.

4. Please include a copy of Table 1 which you refer to in your text on page 16.

Reviewers' comments:

Reviewer's Responses to Questions

**Comments to the Author**

1. Does the manuscript provide a valid rationale for the proposed study, with clearly identified and justified research questions?

Reviewer #1: Yes

Reviewer #2: Yes

2. Is the protocol technically sound and planned in a manner that will lead to a meaningful outcome and allow testing the stated hypotheses?

Reviewer #1: Partly

Reviewer #2: Partly

3. Is the methodology feasible and described in sufficient detail to allow the work to be replicable?

Reviewer #1: No

Reviewer #2: No

4. Have the authors described where all data underlying the findings will be made available when the study is complete?

Reviewer #1: Yes

Reviewer #2: Yes

5. Is the manuscript presented in an intelligible fashion and written in standard English?

Reviewer #1: Yes

Reviewer #2: Yes

6. Review Comments to the Author

You may also provide optional suggestions and comments to authors that they might find helpful in planning their study.

Reviewer #1: A research study is being proposed which aims to explore the utilization of healthcare schemes among tribal reproductive-age groups in Jharkhand India with an ultimate goal of improving access to quality healthcare services. The research will be conducted in two phases. Phase one will assess the knowledge, attitude, and practices of healthcare schemes. Phase two will develop and validate an awareness booklet.

Major revision:

Include a comprehensive statistical analysis plan detailing the analysis to be conducted for each phase of the study.

Minor revisions:

Identify the software that will be used for statistical analysis.

Reviewer #2: 1. In the justification for the study the authors mention Uncontrolled fertility. However, the supporting data is only of contraceptive use. There is no mention of TFR or NRR. Without a high TFR the authors can not comment on fertility.

2. Objectives - The objectives are listed as bullet points. But in the methodology section the authors mention obj 1,2 &3

3. There is no separate section with Description of the Study Setting for the reader to understand the context. We expect population information, tribes, languages spoken, livelihood, terrain, access to health care etc.

4. Study Design & Methodology-

4a. Sequential explanatory design uses qualitative data to explain quantitative data- This is partly done for the phase 1. A note on which aspects will be addressed using this design will help. Authors should clearly mention the quantitative component, the qualitative component and the booklet design and testing component.

4b. What is the age group? 10-19, 15-49 is mentioned, but the revised protocol table only shows age starting from 15

4c. Religion - Why is Sarna not listed

4d. Marital Status- Where do separated women get listed?

5. Questionnaire is Self administered. So the assumption is that all participants are literate. Then why is thumb impression needed?

6. Authors mention that the sampling is simple random sampling and stratified sampling. Which part is simple random sampling?

7. What is the basis for a design effect of 1.5?

8. A simple explanation for using Phenomenology approach must be added.

9. Working with the marginalised community - There is no mention of how the researchers plan to observe Cultural Sensitivity (I see that all authors are non-tribals). It is mentioned that permission from MO will be taken for the field research. Are the authors sure about how to approach to the tribal communities? An approach mentioned in the paper will only give screwed results because the approach is from health sector.

10. Implementation- The paper mentions Implementation. It is confusing. Implementation of what? Is the phase 2 about developing a booklet and testing it, or something else?

Is it content and media development for a booklet or strategy, content, method and media development and trial?

11. How many languages is the booklet going to be in? Which are the main languages spoken?

12. Who will use the booklet and how will it be tested?

7. PLOS authors have the option to publish the peer review history of their article (what does this mean? ). If published, this will include your full peer review and any attached files.

**Do you want your identity to be public for this peer review?** For information about this choice, including consent withdrawal, please see our Privacy Policy .

Reviewer #1: No

Reviewer #2: **Yes: ** Shantidani Minz

---

## [Author Response · Author response to Decision Letter 1]

27 Dec 2024

I am thankful to the valuable comments given by the reviewer.

Kindly requesting for full waive off for if got accepted. As we didn't got any funding till now.

Kind regards

Dr Jarina Begum (Corresponding author) & Dr Rohit Raj (1st author)

---

## [Decision Letter · Decision Letter 1]

7 Mar 2025

PONE-D-24-37230R1Development & Validation of a health awareness booklet: “Reproductive Health Schemes for tribal women of Jharkhand” a study protocolPLOS ONE

Dear Dr. Begum,

Thank you for submitting your manuscript to PLOS ONE. After careful consideration, we feel that it has merit but does not fully meet PLOS ONE’s publication criteria as it currently stands. Therefore, we invite you to submit a revised version of the manuscript that addresses the points raised during the review process.

We look forward to receiving your revised manuscript.

Kind regards,

George Kuryan

Academic Editor

PLOS ONE

Journal Requirements:

Reviewers' comments:

Reviewer's Responses to Questions

**Comments to the Author**

1. Does the manuscript provide a valid rationale for the proposed study, with clearly identified and justified research questions?

Reviewer #2: Yes

2. Is the protocol technically sound and planned in a manner that will lead to a meaningful outcome and allow testing the stated hypotheses?

Reviewer #2: Yes

3. Is the methodology feasible and described in sufficient detail to allow the work to be replicable?

Reviewer #2: Yes

4. Have the authors described where all data underlying the findings will be made available when the study is complete?

Reviewer #2: Yes

5. Is the manuscript presented in an intelligible fashion and written in standard English?

Reviewer #2: Yes

6. Review Comments to the Author

You may also provide optional suggestions and comments to authors that they might find helpful in planning their study.

Reviewer #2: The training involving Nukkad Natak in each village is not part of any objective. It is likely to contaminate the evaluation of the booklet. Consider removing it from the protocol and doing it after the study completion if the institution wants to do it as a service.

7. PLOS authors have the option to publish the peer review history of their article (what does this mean? ). If published, this will include your full peer review and any attached files.

**Do you want your identity to be public for this peer review?** For information about this choice, including consent withdrawal, please see our Privacy Policy .

Reviewer #2: **Yes: ** Shantidani Minz

---

## [Author Response · Author response to Decision Letter 2]

18 Mar 2025

Reference list have been refined and updated.

---

## [Decision Letter · Decision Letter 2]

28 Mar 2025

Development & Validation of a health awareness booklet: “Reproductive Health Schemes for tribal women of Jharkhand” a study protocol

PONE-D-24-37230R2

Dear Dr.Begum

We’re pleased to inform you that your manuscript has been judged scientifically suitable for publication and will be formally accepted for publication once it meets all outstanding technical requirements.

Kind regards,

George Kuryan

Academic Editor

PLOS ONE

Additional Editor Comments (optional):

Reviewers' comments:

Reviewer's Responses to Questions

**Comments to the Author**

1. Does the manuscript provide a valid rationale for the proposed study, with clearly identified and justified research questions?

Reviewer #1: Yes

2. Is the protocol technically sound and planned in a manner that will lead to a meaningful outcome and allow testing the stated hypotheses?

Reviewer #1: Yes

3. Is the methodology feasible and described in sufficient detail to allow the work to be replicable?

Reviewer #1: Yes

4. Have the authors described where all data underlying the findings will be made available when the study is complete?

Reviewer #1: No

5. Is the manuscript presented in an intelligible fashion and written in standard English?

Reviewer #1: Yes

6. Review Comments to the Author

You may also provide optional suggestions and comments to authors that they might find helpful in planning their study.

Reviewer #1: All comments have been adequately addressed.

7. PLOS authors have the option to publish the peer review history of their article (what does this mean? ). If published, this will include your full peer review and any attached files.

**Do you want your identity to be public for this peer review?** For information about this choice, including consent withdrawal, please see our Privacy Policy .

Reviewer #1: No

---

## [Editor Report · Acceptance letter]

PONE-D-24-37230R2

PLOS ONE

Dear Dr. Begum,

I'm pleased to inform you that your manuscript has been deemed suitable for publication in PLOS ONE. Congratulations! Your manuscript is now being handed over to our production team.

Kind regards,

on behalf of

Professor George Kuryan

Academic Editor

PLOS ONE